# Contamination by Antibiotic-Resistant Bacteria on Cell Phones of Vendors in a Peruvian Market

**DOI:** 10.3390/medicina59040669

**Published:** 2023-03-28

**Authors:** Walter Gómez-Gonzales, Anthony Alvarado-Garcia, Marytté Suárez-Mamani, Bernardo Dámaso-Mata, Vicky Panduro-Correa, Jorge L. Maguiña, Samuel Pecho-Silva, Ali A. Rabaan, Alfonso J. Rodriguez-Morales, Kovy Arteaga-Livias

**Affiliations:** 1Escuela de Medicina-Filial Ica, Universidad Privada San Juan Bautista, Ica 11001, Peru; walter.gomez@upsjb.edu.pe; 2Facultad de Medicina, Universidad Nacional Hermilio Valdizán, Huánuco 10003, Peru; anthony_9m@hotmail.com (A.A.-G.); lieblich_20@outlook.es (M.S.-M.); bdamaso@unheval.edu.pe (B.D.-M.); vpanduro@unheval.edu.pe (V.P.-C.); 3Hospital Regional Hermilio Valdizan, Huánuco 15011, Peru; 4Facultad de Ciencias de la Salud, Universidad Científica del Sur, Lima 15067, Peru; jorge.luis.maguina@gmail.com (J.L.M.); arodriguezmo@cientifica.edu.pe (A.J.R.-M.); 5Hospital Nacional Edgardo Rebagliati Martins, Lima 15072, Peru; samuelpechosilva@gmail.com; 6Molecular Diagnostic Laboratory, Johns Hopkins Aramco Healthcare, Dhahran 31311, Saudi Arabia; arabaan@gmail.com; 7College of Medicine, Alfaisal University, Riyadh 11533, Saudi Arabia; 8Department of Public Health and Nutrition, The University of Haripur, Haripur 22610, Pakistan; 9Grupo de Investigación Biomedicina, Faculty of Medicine, Fundación Universitaria Autónoma de las Américas-Institución Universitaria Visión de las Américas, Pereira 660003, Colombia; 10Gilbert and Rose-Marie Chagoury School of Medicine, Lebanese American University, Beirut P.O. Box 36, Lebanon; 11Hospital II EsSalud, Huánuco 10001, Peru

**Keywords:** cell phone, bacterial contamination, antibiotic bacterial resistance, community-acquired infections, *Staphylococcus aureus*

## Abstract

*Background and Objectives.* Multiple studies have evaluated the presence of bacterial contamination on cell phones in clinical settings; however, the presence and transmission of antibiotic-resistant bacteria on cell phones in the community have not been adequately elucidated. *Material and Methods***.** A cross-sectional study was carried out to determine the presence of bacteria resistant to antibiotics on the cell phones of vendors in a Peruvian market and the associated factors. A sample of 127 vendors was obtained through stratified probabilistic sampling using a data collection form validated by experts. Cell phone samples were cultured using a standard technique, and antibiotic sensitivity was determined using the Kirby–Bauer technique. Chi-squared and Mann-Whitney U tests were used to determine factors associated with resistance in cell phone cultures. *Results***.** Among the cell phones, 92.1% showed bacterial growth, predominantly Gram-positive bacteria (coagulase-negative staphylococci and *Staphylococcus aureus*), and 17% of the cultures showed resistance to at least three antibiotics evaluated. Two strains fell into the category of methicillin-resistant *S. aureus*, and three strains of *E. coli* had resistance to carbapenems. *Conclusions***.** A short distance between customers and vendors, lack of a cell phone case, and having a cell phone with touchscreen are factors associated with antibiotic-resistant bacteria on cell phones.

## 1. Introduction

Today, cell phones are essential accessories in people’s lives. In 2020, there were more than 7.5 billion cell phones worldwide, exceeding the total population [1]. Due to the range of functions of these devices, they are used at work [2,3], in kitchens, markets, and even in bathrooms [4]. Cell phones are used to make calls, send text messages, and obtain information, and due to their high frequency of use, they harbor microorganisms such as bacteria, fungi, or viruses [5]. Since microorganisms are present on all surfaces, contact with any object can lead to their transmission [6,7].

Since the first study describing the presence of bacteria on cell phones in 2007, it has been determined that continuous use, temperatures between 25 °C and 43 °C, and the humidity produced by these devices greatly favor the growth of bacteria [8,9,10]. A cell phone can harbor more microorganisms than a man’s toilet seat, the bottom of a shoe, or a doorknob [11]. With the advent of molecular biology technologies and the study of the microbiome in these devices, our knowledge of pathogenic and resistant bacteria that contaminate cell phones will likely develop [12,13].

Several studies have shown that disinfection of the surface of cell phones with isopropyl alcohol or even cleaning with wet cotton can reduce the presence of bacteria by up to 70–80% [5,14]. All the disinfection, cleaning, and education measures that were implemented by the users decreased the bacterial load of cell phones immediately and even for up to 12 months after the intervention [15,16]. Unfortunately, up to 95% of people never disinfect their cell phones [8].

Multiple studies have assessed the presence of bacterial contamination on cell phone surfaces, especially in healthcare and university contexts; however, the possibility for cell phones to carry and can transmit pathogenic or antibiotic-resistant bacteria in community settings remains to be elucidated [4,17]. In a study including 400 cell phones, Akinyemi et al. found the highest proportion of contamination (37%) among food vendors, followed by students, public servants, and health personnel [18].

Several investigations indicate that the transmission of bacteria, in general, and strains resistant to antibiotics can occur through the consumption of contaminated food, contact with feces or secretions of animals, or direct contact with people who carry the bacteria [19,20]. Since they meet all the above-mentioned requirements, food vendors at markets act as facilitators for transmitting bacteria to the community [21,22].

The microbiota present in the cell phones of people who work outside of hospital settings may be very different from that found among health workers. This is probably due to lower biosecurity, disinfection, and cleaning measures in the community, as well as different measures of introducing microbiological pressure, such as using antimicrobials in the livestock and poultry industries [5]. In this context, resistant bacteria can be found with the same or even greater frequency than in healthcare on cell phones from people who work at and visit markets [23].

Bacteria can generate antibiotic resistance in both healthcare and community settings [24]. Therefore, the objective of this study was to determine the presence of cell phone contamination by antibiotic-resistant bacteria and the factors related to this contamination among vendors in the main market of the city of Huánuco, Peru.

## 2. Materials and Methods

### 2.1. Study Design

An analytical cross-sectional study was conducted in the main market of the city of Huánuco, Peru, from November to December 2020.

### 2.2. Population and Sample

The population studied were the vendors of the main market of the city of Huanuco, which serves approximately 235,529 inhabitants. Although the market structure is covered, the roof is more than 20 m high. The 1.5 m long stalls are adjacent and share two entrances/exits at the ends of the row of 25 stalls. Data collection was conducted in the spring season. However, it is important to note that in Huanuco, the city where the study was conducted, the climate is temperate, and there is no notable variation between seasons. The study population consisted of 214 vendors with cell phones, who belonged to sectors where the main activity was selling food (cooked or prepared), meat, fruit or vegetables, and groceries. Stratified probabilistic sampling was performed using the Epidat v.3.1 program, with a confidence level of 95%, obtaining a sample of 127, stratified according to Figure 1.

### 2.3. Variables and Instruments

The study utilized a structured questionnaire as the research instrument. The questionnaire consisted of 6 sections and 27 open-ended questions that covered various aspects of the research topic. These sections were: (1) sociodemographic characteristics: age, sex, sector of work, and working hours, (2) customer contact, customer permanence in minutes and customer distance in meters, (3) cell phone use: approximate use, bathroom use, cell phone sharing, (4) disinfection practices, (5) perception of cell phone contamination, and (6) cell phone characteristics: use and material of screen protector, use and material of equipment protector, cracks, and type of cell phone. The questionnaire was validated based on the judgment of 5 experts in infectious diseases and epidemiology with a Cronbach alpha reliability value of 81%. The variable resistance was defined as the presence of antimicrobial resistance to at least one antibiotic evaluated.

### 2.4. Isolation and Sensitivity of Bacteria

The samples were collected by clinical laboratorians with expertise in sample collection. Hands were adequately washed with an alcohol-based hand sanitizer prior to sampling, and powder-free disposable gloves and masks were worn for each sample throughout the work process to avoid cross-contamination. A mobile phone swab sample was collected from each participant. A sterile cotton swab soaked in sterile saline solution was rotated by sliding it over the entire area of the phone (screen, keyboard, sides, and back). For phones with a protective case, the sample was collected from the outer surface of the case in addition to the screen. The swab was immediately placed in a test tube with 1 mL of 0.9% physiological saline solution for transportation. The duration of the transport was less than 1 h, keeping in transport boxes with a cold chain, such as the one used in the transport of vaccines. In the laboratory, each sample was cultured for 24 h in a Petri dish containing 20 mL of blood agar. The colonies that grew on blood agar were stained with Gram staining to subsequently seed the colony on MacConkey agar (for Gram-negatives) and Mannitol salt agar (for Gram-positives) at a temperature of 37 °C. After 48 h, different biochemical tests were used for correct identification of the species, such as iron triple sugar agar, indole, citrate, oxidase, urease, motility, methyl red, mannitol, catalase, and coagulase, according to previous studies [25,26]. The sensitivity of microorganisms to 13 different antibiotics was tested. For Gram-positive bacteria, the antibiotics tested were clindamycin, vancomycin, azithromycin, and erythromycin. For Gram-negative strains, the antibiotics tested were fosfomycin, cefoxitin, imipenem, meropenem, ampicillin-sulbactam, cefazolin, levofloxacin, colistin, and cefotaxime. The test was determined by the Kirby–Bauer disk diffusion technique in Mueller–Hinton agar according to the guidelines of the Clinical and Laboratory Standards Institute [27]. Briefly, pure isolates (4–5 colonies) were added to sterile tubes containing 5 mL of saline and mixed gently until a homogeneous suspension was formed. The bacterial suspension’s turbidity was uniform using a 0.5 McFarland standard. A sterile cotton swab was dipped into the suspension and inoculated the bacterial suspension over the entire surface of Mueller–Hinton agar (Oxoid Ltd., Basingstoke, Hampshire, UK), which was left at room temperature to dry for 3–5 min [26]. Due to the standardization and wide use of microbiological tests, as well as limited resources, no reprocessing or parallel testing was performed. *E. coli* (ATCC 25922) was used as an internal control strain in all performed tests.

### 2.5. Ethical Aspects

The Ethics Committee of the University Research Directorate of the Universidad Nacional Hermilio Valdizán approved the research. Written informed consent was obtained from all subjects before inclusion. Each participant was informed of the risks and benefits of the procedure before signing the informed consent. The data of all participants were confidential.

### 2.6. Statistical Analysis

After data collection, a spreadsheet was created in Microsoft Excel v. Windows 2019, in which the data were verified, before we performed statistical tests using Stata 16.0 software (StataCorp, 4905 Lakeway Dr, College Station, TX, USA). Descriptive statistical analysis of the information was carried out through frequencies, percentages, and measures of central tendency (mean, median, mode). For the bivariate inferential analysis, the Mann–Whitney U test was used for quantitative variables, and the chi-squared test was used for the association between qualitative variables. A value of *p* < 0.05 with a 95% confidence interval was considered statistically significant.

## 3. Results

The study population had a mean age of 43 years, and females predominated. A high proportion of people were aware of the presence of bacteria on cell phones (59.1%), and most of the phones were touchscreen (88.9%) (Table 1).

Bacterial growth was detected on the surface of 117 (92.1%) of the cell phones, with the highest proportion of bacteria isolated being Gram-positive (82%). All the isolates had growth above 10,000 CFU. Among the positive cultures, 65.8% presented resistance to at least one antibiotic evaluated.

The bivariate analysis results revealed a significant correlation between the presence of antibiotic-resistant bacteria on cell phones and the usage of a device protector, the type of protector, and the cell phone model (Table 2).

The bacteria identified were *S. aureus*, Coagulase-negative staphylococci, *Escherichia coli*, *Enterococcus faecalis*, *Enterobacter aerogenes*, *Serratia marcescens*, and *Enterobacter cloacae*. Some strains of the Coagulase-negative staphylococci presented resistance to up to five antibiotics, and strains of *S. aureus* and *E. coli* bacteria presented resistance to up to four antibiotics. Two strains fell into the category of methicillin-resistant *S. aureus* and three strains of *E. coli* had resistance to carbapenemics (Table 3 and Table 4).

## 4. Discussion

The combination of constant handling and the heat generated by cell phones provides an excellent breeding ground for all kinds of microorganisms, especially those found on the owner’s skin [28]. Indeed, one interesting study reported that 82% of the participants’ skin microbiota was transmitted to their mobile phones’ screens [29].

Bacterial resistance is one of the great problems of our century. The COVID-19 pandemic caused an increase in the prevalence of antibiotic-resistant bacteria, probably due to the increased use of antibiotics, particularly carbapenems, as one of its many causes [30,31]. The presence of multidrug-resistant bacteria in community contexts is increasingly frequent and requires maximum attention [32]. Despite the general understanding that cell phones used by healthcare workers are more likely to harbor bacteria on their surfaces due to their exposure to contaminated environments [33], some studies have revealed that cell phone bacteria are actually more prevalent among people in community settings [4,18], perhaps due to increased use or poorer device cleaning and hygiene measures.

Urban markets serve as potential hotspots for the emergence and dissemination of infectious diseases as numerous factors converge there. The recent COVID-19 pandemic, originating from a market in Wuhan, serves as a poignant illustration of this phenomenon [19,34]. Numerous studies have documented the presence of antibiotic-resistant bacteria in products such as meat, fish, shellfish, and dairy. These bacteria may also be present on common market tools, such as scales, slicers, refrigerators, and, of course, cell phones used by vendors [21,35,36,37].

The present study found that the disinfection of a cell phone was not related to the presence of positive cultures. Although disinfection and cleaning habits are known to help prevent the spread of disease, there have been some conflicting results regarding their benefits on cell phones. Multiple studies have shown the relationship between disinfection and bacterial contamination in cell phones. Gashaw et al. found that 70% alcohol produced a significant reduction in the rate of mobile phone contamination (*p* < 0.0001) [38]. Another study reported that disinfecting with 70% isopropyl alcohol reduced contamination [39]. Thus, routine daily cleaning protocols favor a decrease in bacterial load, making transmission less likely [40]. On the contrary, Martina et al. described that although alcohol gel is an effective disinfection method, it does not provide as complete disinfection as expected [41]. These results suggest that bacteria on cell phones is inversely related to the frequency of disinfection, with researchers less likely to find a bacterial presence on cell phones disinfected more frequently.

At the same time, the cleaning habits of the population changed with the COVID-19 pandemic. While previous studies showed that a small proportion of people cleaned their cell phones [42,43], this has now been expanded, reducing the proportion of people who currently do not disinfect or at least clean their cell phone [44,45]. Increased concerns about personal hygiene and the spread of disease that have led to this change may also produce benefits concerning better care, cleaning, and disinfection of other personal items.

An important finding is that distance from the customer protects against cell phone contamination. Although no study has evaluated the effect of the distance between people on avoidance of cell phone contamination, several studies have shown the effectiveness of this measure in protecting against contagion by the SARS-CoV-2 virus [46,47]. Indeed, it is likely that individuals who maintain a greater distance from people for protective purposes against SARS-CoV-2, according to Peruvian government measures, are also more attentive to the hygiene of their cell phone. Another factor to be studied is where the cell phone is kept or placed during customer service, with cross-contamination being more likely when the device is outdoors.

A statistically significant reduction in the frequency of antibiotic-resistant bacteria was found in cell phones with equipment protectors. An interesting study in China showed that the use of plastic protectors for cell phones tended to reduce the presence of bacterial contamination. The authors suggested that plastic covers allow better disinfection of cell phones, and bacterial colonies do not adhere as easily to these covers as they do to cell phones [48]. A community study by Bhoonderowa et al. found that the use of cell phone protectors decreased the bacterial load [49], and Bodena et al. reported that cell phones without a protective cover were at greater risk of bacterial contamination [26]. On the contrary, Jansen et al. described how fabric or leather covers favored the presence of positive cultures on cell phones [1], while Cicciarella Modica et al. indicated that flexible protectors conditioned the presence of a higher bacterial load [42]. Currently, many protective surfaces are smoother than mobile phone screens, which may lower bacterial contamination [50]; however, Shakir et al. stated that there was no difference in bacterial growth between smooth and rough covers [51]. Another factor that could account for the varied outcomes is the attitude toward the cover, leading to differing levels of cleanliness and disinfection among owners. Given these conflicting results, more research is needed to fully understand the effect that different cell phone covers have on bacterial growth.

This study shows that cell phones with keys have less contamination than those with touchscreen. This result is striking since Dorost et al. showed that cell phones with a keypad had a higher frequency of contamination [52], and Pal et al. reported that cell phones with a touchscreen were less colonized than those with a keypad [50]. A possible explanation could be the availability of many applications. Users tend to spend more time touching the surface of cell phones with touchscreen than those with a keypad, leading to greater contamination, as described by Lee et al. [53]. This is reinforced by a study performed in 2014 which showed that, on average, a cell phone was touched up to 150 times a day [39].

Multiple studies have documented the presence of antibiotic-resistant bacteria on cell phones in various settings, including healthcare facilities and universities [1,5,24,38,39]. Our research has uncovered a notably high prevalence of antibiotic resistance among the cellphones owned by market vendors. This underscores the magnitude of the issue and emphasizes the importance of tackling it beyond just hospital settings. Furthermore, additional studies have identified the presence of multidrug-resistant enterobacteria and even extended-spectrum beta-lactamases on both meat products and cell phones [21,22,41].

We observed that a high rate of bacteria had resistance to more than two antibiotics, with up to five strains of Coagulase-negative staphylococci showing resistance to five antibiotics. Martina et al. found that Gram-negative bacteria isolated from cell phones presented a high percentage of resistance compared to Gram-positive bacteria [41], while Gashaw et al. reported that 17% of cell phone cultures were resistant to at least two antimicrobials and that, similar to the study by Al Momani, some bacterial strains were resistant to up to six antibiotics, suggesting that this could pose a major threat to public health [38,54].

Carbapenem resistance is a growing problem in public health. Recently, it has been observed that certain contaminating bacteria present on cell phones and medical uniforms may carry resistance genes to these important antibiotics. This is of particular concern because carbapenems are one of the last therapeutic options for treating bacterial infections resistant to other antibiotics.

One of the primary limitations of our study is the lack of a comprehensive history of coexisting medical conditions experienced by the participants. However, it is reasonable to assume that individuals with chronic conditions requiring frequent visits to healthcare centers would need help to perform their duties effectively in the demanding work environment of a market setting. Another significant limitation of this study is the lack of parallel testing due to limited resources. Consequently, some of the identified antimicrobial profiles may have been affected by the misidentification of organisms or problems with the Kirby–Bauer technique. In particular, the high incidence of vancomycin-resistant *S. aureus* among the 18 reported isolates, which is a rare phenomenon, and the pattern observed in coagulase-negative staphylococcal species are highly unusual and suggest possible problems with the accuracy of the data.

## 5. Conclusions

In conclusion, our study highlights several key factors associated with the presence of antibiotic-resistant bacteria on cell phones, including proximity to customers, lack of protective covers, and the use of touchscreen devices. The contamination of cell phones with pathogenic and antibiotic-resistant bacteria extends beyond healthcare settings and poses a significant threat to the control of antibiotic resistance in the wider community. Our findings underscore the importance of implementing measures to prevent the spread of resistant bacteria and promote good hygiene practices among cell phone users.

## Figures and Tables

**Figure 1 medicina-59-00669-f001:**
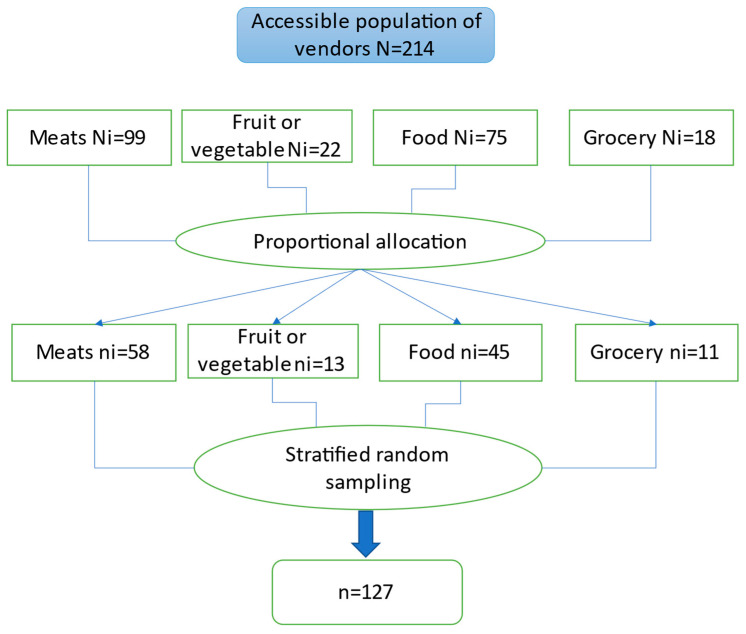
Determination of the sample size of vendors. N = Total size of the accessible population. Ni = Size of the population of each sector. ni = Sample size drawn from each sector. n = Required sample size.

**Table 1 medicina-59-00669-t001:** Demographic and cell phone use characteristics of the vendors in the Huánuco market, 2020 (n = 127).

Variable	Frequency	Percentage
**Sex**		
**Female**	94	74.1
**Male**	33	25.9
**Age**		
**(X + SD)**	43.4 + 12.9	
**Sector**		
**Foods**	45	35.4
**Fruits and vegetables**	13	10.2
**Grocery**	11	8.6
**Meats**	58	45.7
**Clients in 1 h**		
**(Median: IQR)**	5: (4–10)	
**Client permanence (minutes)**		
**(Median: IQR)**	5: (4–10)	
**Client distance (meters)**		
**(X + SD)**	1.3 + 0.3	
**Knowledge of bacteria on cell phones**		
**Yes**	75	59.1
**No**	52	40.9
**Share cell phone with others**		
**No**	115	90.6
**Yes**	12	9.4
**Use of cell phone in the bathroom**		
**No**	94	74.1
**Yes**	33	25.9
**Cell phone disinfection**		
**No**	14	11.1
**Yes**	113	88.9
**Use of screen protector**		
**No**	32	25.2
**Yes**	95	74.8
**Screen protector type**		
**Plastic**	13	13.7
**Fiberglass**	82	86.3
**Use of equipment protector**		
**No**	49	38.6
**Yes**	78	61.4
**Equipment protector type**		
**Plastic**	68	87.2
**Others**	10	12.8
**Cell phone type**		
**Touchscreen**	113	88.9
**Keys**	14	11.1
**Cracked screen**		
**No**	76	59.8
**Yes**	51	40.2

X: Mean. SD: Standard deviation. IQR: Interquartile range.

**Table 2 medicina-59-00669-t002:** Bivariate analysis showing the presence of antibiotic-resistant bacteria in cell phones from market vendors in the Huánuco market, 2020.

Characteristics	Antibiotic Resistance				*p ^#^*
	Negative	%	Positive	%	
**Sex**					
**Female**	30	34.9	56	65.1	0.792
**Male**	10	32.3	21	67.7	
**Age**					
**(X + SD)**	43.8 + 12.7		42.7 + 13.1		0.709 *
**Sector**					0.148
**Foods**	17	41.5	24	58.5	
**Fruits and vegetables**	5	35.5	8	61.5	
**Grocery**	13	24.1	41	75.9	
**Meats**	5	55.6	4	44.4	
**Clients in 1 h**					0.921 *
**(Median: IQR)**	5: (4–10)		6: (4–10)		
**Client permanence (minutes)**					0.998 *
**(Median: IQR)**	5: (4–10)		5: (4–10)		
**Client distance (meters)**					0.023 *
**(X + SD)**	1.45 + 0.27		1.29 + 0.39		
**Knowledge of bacteria in cell phones**					0.666
**Yes**	16	32	34	68	
**No**	24	35.8	43	64.2	
**Share cell phones with others**					0.909
**No**	30	34.5	57	65.5	
**Yes**	10	33.3	20	66.7	
**Use of cell phone in the bathroom**					0.503
**No**	16	20	64	80	
**Yes**	4	14.3	24	85.7	
**Cell phone disinfection**					0.13
**No**	2	15.4	11	84.6	
**Yes**	38	36.5	66	63.5	
**Use of screen protector**					0.133
**No**	14	45.2	17	54.8	
**Yes**	26	30.2	60	69.8	
**Screen protector type**					0.351
**Plastic**	2	18.2	9	81.8	
**Fiberglass**	24	32	51	68	
**Use of equipment protector**					0.012
**No**	22	47.8	24	52.2	
**Yes**	18	25.4	53	74.6	
**Equipment protector type**					0.026
**Plastic**	13	20.9	49	79.1	
**Others**	5	55.6	4	44.4	
**Cell phone type**					0.005
**Touchscreen**	31	29.8	73	70.2	
**Keys**	9	69.2	4	30.7	
**Cracked screen**					0.223
**No**	26	38.8	41	61.2	
**Yes**	14	28	36	72	

#: Chi-squared. *: Mann–Whitney U. X: Mean. SD: Standard deviation. IQR: Interquartile range.

**Table 3 medicina-59-00669-t003:** Microorganisms isolated and their sensitivity pattern (n = 117).

Bacteria\Antibiotics	CLI			VAN			AZM			ERI			FOS			FOX			IMP			MEM			SAM			CFZ			LVX			COL			CTX		
	R	I	S	R	I	S	R	I	S	R	I	S	R	I	S	R	I	S	R	I	S	R	I	S	R	I	S	R	I	S	R	I	S	R	I	S	R	I	S
** *S. aureus* **	4	3	11	5	2	11	2	10	6	4	12	2	5	2	11	2	10	6	NR	NR	NR	NR	NR	NR	NR	NR	NR	NR	NR	NR	NR	NR	NR	NR	NR	NR	NR	NR	NR
**(total = 18)**																																							
**Coagulase-negative staphylococci**	17	21	23	12	9	40	15	23	23	10	37	14	23	16	22	13	44	4	NR	NR	NR	NR	NR	NR	NR	NR	NR	NR	NR	NR	NR	NR	NR	NR	NR	NR	NR	NR	NR
**(total = 61)**																																							
** *E. coli* **	NR	NR	NR	NR	NR	NR	NR	NR	NR	NR	NR	NR	NR	NR	NR	NR	NR	NR	0	3	11	3	5	6	2	7	5	4	10	0	4	5	5	4	1	9	4	9	1
**(total = 14)**																																							
** *E. faecalis* **	NR	NR	NR	4	9	4	3	7	7	3	13	1	NR	NR	NR	2	8	7	NR	NR	NR	NR	NR	NR	1	2	14	NR	NR	NR	2	6	9	NR	NR	NR	NR	NR	NR
**(total = 17)**																																							
** *E. aerogenes* **	NR	NR	NR	NR	NR	NR	NR	NR	NR	NR	NR	NR	NR	NR	NR	NR	NR	NR	0	0	4	0	0	4	1	2	1	2	2	0	0	0	4	0	2	2	0	4	0
**(total = 4)**																																							
** *S. marcescens* **	NR	NR	NR	NR	NR	NR	NR	NR	NR	NR	NR	NR	NR	NR	NR	NR	NR	NR	0	0	2	0	0	2	0	0	2	0	2	0	0	0	2	0	0	2	0	2	0
**(total = 2)**																																							
** *E. cloacae* **	NR	NR	NR	NR	NR	NR	NR	NR	NR	NR	NR	NR	NR	NR	NR	NR	NR	NR	1	0	0	0	0	1	0	1	0	0	1	0	0	0	1	0	0	1	0	0	1
**(total = 1)**																																							

NR: Not realized. CLI: Clindamycin. VAN: Vancomycin. AZM: Azithromycin. ERI: Erythromycin. FOS: Fosfomycin. FOX: Cefoxitin. IPM: Imipenem. MEM: Meropenem. SAM: Ampicillin–Sulbactam. CFZ: Cefazolin. LVX: Levofloxacin. COL: Colistin. CTX: Cefotaxime.

**Table 4 medicina-59-00669-t004:** Frequency of antibiotic resistance (n = 117).

Bacteria	Frequency of Antibiotic Resistance						Total
	Sensitive	Resistant to 1 ATB	Resistant to 2 ATB	Resistant to 3 ATB	Resistant to 4 ATB	Resistant to 5 ATB	
** *Staphylococcus aureus* **	7 (38.9)	4 (22.2)	4 (22.2)	2 (11.1)	1 (5.6)		18
**Coagulase-negative staphylococci**	20 (32.8)	19 (31.2)	9 (14.8)	5 (8.2)	3 (4.9)	5 (8.2)	61
** *Enterococcus faecalis* **	5 (29.4)	9 (52.9)	3 (17.7)				17
** *Escherichia coli* **	5 (35.7)	4 (28.6)	1 (7.1)	1 (7.1)	3 (21.4)		14
** *Enterobacter aerogenes* **	2 (50)	1 (25)	1 (25)				4
** *Enterobacter cloacae* **		1 (100)					1
** *Serratia marcescens* **	1 (50)	1 (50)					2
	40	39	18	8	7	5	117
**ATB = Antibiotics**							

## Data Availability

Not applicable.

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
