# Peer review of "Contamination by Antibiotic-Resistant Bacteria on Cell Phones of Vendors in a Peruvian Market"

_medicina, 2023, doi:10.3390/medicina59040669_

Round 1

Reviewer 1 Report

The manuscript provides a cross-sectional study to determine the presence of bacteria resistant to antibiotics on the cell phones of vendors in a Peruvian market and the associated factors. For this, A sample of 127 vendors was obtained through stratified probabilistic sampling using a data collection form validated by experts. The manuscript highlights the critical factors associated with antibiotic-resistant bacteria on cell phones, including proximity to customers, lack of protective covers, and touchscreen devices. The authors described the methodology and results very well. Overall, the manuscript is very well written, the Authors properly conducted all the experiments, and an exciting approach is presented. I think publishing this manuscript in its original form would be a good idea. However, I have some queries as follows:

1.              Many studies are already available on the prevalence of antibiotic-resistant bacteria on the surface of equipment.  Authors should explain their newness and how it differs from previous studies.

2.              Authors should clarify in the methodology section how they maintained an aseptic condition during the collection of samples. The samples may be contaminated at the time of collection.

3.              What are the selection criteria for choosing these antibiotics for the antibiotic profiling experiment?

Reviewer 2 Report

I have reviewed the manuscript “Contamination by antibiotic-resistant bacteria in cell phones of vendors in a Peruvian market." submitted to “Medicina” for publication. In this study, samples were taken from the mobile phones of 127 volunteers to check the bacterial contamination. The results of this study show that hospital-resistant strains such as MRSA and CR-E. coli were isolated from phones, which can indicate the spread of these bacteria by phones in the community or the presence of carriers. In my opinion, the results of this research can be interesting for the reader. However, the authors should point out in the discussion section the importance of the spread of resistant strains in the community by mobile phones.

Comment to the authors,

Line 125-128: Please mention the names of antibiotics used separately for Gram-positive and Gram-negative bacteria.

Line 128 and table3 and 4: The report of antibiotic susceptibility of colistin with Kirby-Bauer disk diffusion technique is not acceptable. The sensitivity and resistance of this antibiotic should be reported by the broth dilution (MIC) method. Please correct it.

Line 136: It must be mentioned which strain (or strains) was used as the positive control of the antibiogram.

Table 3: Three carbapenem resistant E. coli isolates were isolated from the phones of which people? (These may be the faecal carriers of this bacteria).

In general, the percentage of MDR and XDR should be mentioned in the results section.

Have any of the participants been members of the medical staff or have they had a history of hospitalization??

The percentage of MRSA should be reported in the results section.

In the discussion section, talk more about two MRSA and three carbapenem resistant E. coli, for example, these strains have been isolated from the phone of which people? Are there medical staff in the family of those who have been isolated from these strains?

It is suggested to mention MRSA and carbapenem resistant E. coli in the abstract section.

Reviewer 3 Report

Thank you for the opportunity to help review the following manuscript: “Contamination by antibiotic-resistant bacteria in cell phones of vendors in a Peruvian market”. Gomez-Gonzales, et al describe the presence of bacteria and their associated antimicrobial profiles on cell phones from vendors in a specific Peruvian market. They also detail the factors associated with a higher burden of antibiotic resistant bacteria. The study is well- designed and interesting. I have a few comments and suggestions for the authors.

Line 44- has it been documented that bacteria actually grow on cell phone surfaces?

Line 160- 10,000 CFU per how much sample? 1 mL? per phone?

Results and Table 3 and 4: I understand the limitation in resources and that parallel testing could not be performed. Some of the antimicrobial profiles are rare and might indicate that the identification of the organism was incorrect or that there may be an issue with the Kirby Bauer technique. For example, vancomycin resistant S. aureus are encountered rarely, yet 7 of the reported 18 S. aureus isolates were non-susceptible to vancomycin. This is highly unusual. The same is true for the coagulase negative staph species. Here is a good resource describing the worldwide prevalence of VISA/VRSA: Shariati A, et al Global prevalence and distribution of vancomycin resistant, vancomycin intermediate and heterogeneously vancomycin intermediate Staphylococcus aureus clinical isolates: a systematic review and meta-analysis. Sci Rep. doi: 10.1038/s41598-020-69058-z.  

Results and Table 4- It may be more informative to present the frequency of antibiotic resistance by classes of antibiotics instead of (or in addition to) individual antibiotics. For example, because of the hierarchy of cephalosporins, one would expect that the E. coli isolates listed that were resistant to cefotaxime (3rd generation cephalosporin) would also be resistant to cefazolin (1st generation). Those E.coli isolates would have resistance in one class (cephalosporins) of antibiotics. Additionally, those isolates that tested resistance to any of carbapenems, may be considered carbapenem -resistant Enterobacterales and describing as such, may indicate more of concern for public health and transmission. These findings listed in the table could be described in more detail in the results and discussion- especially since carbapenems were mentioned by the authors in line 185.

very minor - Line 260-266  and throughout manuscript- “Gram” and “coagulase” do not need to be capitalized

Reviewer 4 Report

1. Author can elaborate on how temperature affects these bacteria contamination as they have mentioned that this study took place in spring season. So, if some points can be mentioned for other seasons, it would be better.

2. They can give a few reasons as to why only vendors were used for the experiment and not any others.

3. Discuss briefly the possible hazards that these microbes may cause along with health issues caused by them.

4. And finally, they should also talk about what alternatives can be undertaken in order to limit such microbial contamination.

5. Are there any other (sustainable) substitutes for plastic covers to reduce the contamination?
